# CMR-net: A cross modality reconstruction network for multi-modality remote sensing classification

Huiqing Wang[1,2]*, Huajun Wang[1], Lingfeng Wu[1]

1 School of Geophysics, Chengdu University of Technology, Chengdu, Sichuan, China, 2 Center for Information and Educational Technology, Southwest Medical University, Luzhou, Sichuan, China

* whq@swmu.edu.cn

## Abstract

In recent years, the classification and identification of surface materials on earth have emerged as fundamental yet challenging research topics in the fields of geoscience and remote sensing (RS). The classification of multi-modality RS data still poses certain challenges, despite the notable advancements achieved by deep learning technology in RS image classification. In this work, a deep learning architecture based on convolutional neural network (CNN) is proposed for the classification of multimodal RS image data. The network structure introduces a cross modality reconstruction (CMR) module in the multi-modality feature fusion stage, called CMR-Net. In other words, CMR-Net is based on CNN network structure. In the feature fusion stage, a plug-and-play module for cross-modal fusion reconstruction is designed to compactly integrate features extracted from multiple modalities of remote sensing data, enabling effective information exchange and feature integration. In addition, to validate the proposed scheme, extensive experiments were conducted on two multi-modality RS datasets, namely the Houston2013 dataset consisting of hyperspectral (HS) and light detection and ranging (LiDAR) data, as well as the Berlin dataset comprising HS and synthetic aperture radar (SAR) data. The results demonstrate the effectiveness and superiority of our proposed CMR-Net compared to several state-of-the-art methods for multi-modality RS data classification.

## Introduction

Hyperspectral (HS) image, multispectral (MS) image, LiDAR image, and synthetic aperture radar (SAR) image are widely employed in RS to capture distinct characteristics of the same ground features. HS image has rich spectral characteristics and can distinguish ground objects with similar textures and different spectra. The MS images effectively capture and represent the color, brightness, and distinctive features of various ground objects, thereby exhibiting remarkable recognition capabilities in urban environments encompassing streets, buildings, water bodies, soil compositions, and vegetation types. The SAR images primarily capture two types of ground target characteristics: structural attributes (such as texture and geometry) and electromagnetic scattering properties (including dielectric and polarization features). With the advancement of aerospace technology, RS has emerged as a pivotal tool in Earth observation.

**Data Availability Statement:** The Houston2013 data can be through http://dase.grss-ieee.org. The Berlin data can be obtained from http://doi.org/10.5880/enmap.2016.002. The data in the manuscript is confirmed to have no special access, other

researchers can access the data in the same way as the authors, and there are no specific dataset titles and/or login numbers required to access the data at any of the listed links.

**Funding:** This article research results by the southwest university of medical sciences field scientific research subject (fund no. 20/00031616, 20/01041224, 20/01041306) support. Grantee Lingfeng Wu was involved in the preliminary research design and data collection in the study. Grantee Huajun Wang participated in the preliminary research design and data collection in the study.

**Competing interests:** The authors have declared that no competing interests exist.

The interpretation of RS images holds immense significance across various research and application domains, encompassing land cover classification [1], target detection, semantic segmentation [2], environmental monitoring, mineral exploration and diagnosis [3], precision agriculture, food security [4] and beyond. Notably, land cover classification stands out as a particularly prominent field of application.

Over the past decade or so, RS data classification has achieved significant success in extracting discriminative features and designing efficient classifiers, However, these classification techniques, both unsupervised and supervised, are currently limited to a single modality. For example, Wu et al. [5] proposed a method based on spatial-frequency channel features Multi-spectral RS data is useful for geospatial object detection. They capture spatial and frequency information and help to improve the accuracy and reliability of object detection tasks in multi-spectral data analysis. Huang et al. [6] proposed a method to extract geometric representations from LiDAR data for point cloud classification using local manifold learning. This approach exploits local manifold learning and aims to extract discriminative geometric features and representations from LiDAR data. These features can then be used for point cloud classification to accurately identify and classify objects in the point cloud. The identification of the structural type of surface materials solely based on a single type of information (e.g., spectral data) poses challenges in terms of accuracy and reliability.

However, RS data obtained using a single sensor has limited ability to recognize substances on the Earth's surface and lacks rich and diverse information. In theory, the utilization of complementary information from multi-channel RS images have the potential to enhance feature classification accuracy and compensate for limitations associated with single-channel imagery. By integrating these data from different modalities, we can obtain more detailed information to construct the entire feature representation [7]. Therefore, diversification of RS data can provide rich spectral and positional information on features to more accurately identify or classify substances on the Earth's surface. The utilization of multi-channel RS data, such as hyperspectral and LiDAR data, enables the acquisition of abundant feature information, which confers a significant advantage over single-modal data. SAR images have different imaging mechanisms and can penetrate clouds, they can be well used as auxiliary data to solve the problem of missing information caused by cloud coverage of optical images [8]. However, it also presents a formidable challenge for developing and investigating multimodal data processing and analysis methods. In recent years, some researchers have proposed some traditional multi-channel algorithms, but the classification accuracy and effect are yet to be improved. With the use and advantages of deep learning networks in RS image classification, researchers have proposed various multimodal RS data classification methods, such as convolutional neural networks (CNNs), recurrent neural networks (RNNs) [9], graph convolutional networks (GCNs) [10], and deep similarity networks [11]. These methods have been utilized for RS image classification, surpassing the performance bottleneck of using single channels and achieving better results [12]. The multichannel data fusion strategy plays a pivotal role in determining the performance of the multichannel network in classification tasks within RS data. These deep learning methods for fusion of multi-modal RS data sources have been proved to be feasible and effective. Therefore, the utilization of multi-channel RS image classification holds significant research value.

In the research process of multi-modal RS image classification, various RS imagery techniques possess the capability to capture a wide array of the Earth's surface, encompassing spectral radiance, reflectance, height information, textural structure, and spatial attributes. For example, the spectral difference between trees on the ground and trees on the roof may not be large, but the height information provided by LiDAR or SAR data can effectively distinguish between the two. In target detection, such as automobile identification, hyperspectral data

have strong spectral discrimination features, however, RGB and MS data can provide rich spatial geographic information. It can be seen that the fusion of multiple RS data sources can achieve the classification of complex scenes, which cannot be achieved using a single data source. Therefore, it is of great practical significance to study multi-modal RS image classification methods.

Despite the recent success of deep learning networks in extracting discriminative features from multi-source RS data, current methods are limited in handling heterogeneous and multi-modal data such as HS and SAR or HS and LiDAR. While these newly developed techniques have proven effective for fusing multiple RS sources, further advancements are needed to address the challenges posed by diverse data types. This may be due to the lack of more advanced fusion strategies to better eliminate the differences between different modalities and obtain effective discriminative features. It is worth noting that the fusion strategy is a key factor in determining the performance of multimodal networks. In order to obtain better classification results of multi-modal RS data, the feature representation of multi-modal RS data is improved, and more compact information fusion and more effective information transmission are realized. In this paper, we propose a multi-modal RS data classification network based on CNN feature extraction and cross-modal fusion reconstruction, called CMR-Net. That is, a new Cross-modal Fusion Reconstruction (CMR) module was introduced, and then it was integrated into the framework network structure based on CNN to realize the classification of multi-modal RS images. In addition, the CMR module can be flexibly embedded into other network framework structures, and the features of different modalities can be reconstructed in a cross-channel manner, so as to obtain more compact feature representations from RS data of different modalities and realize effective information transmission. The specific contributions of this paper are as follows:

1. In this paper, we propose a convolutional neural network (CNN)-based architecture for the classification of multimodal RS image data, which introduces a CMR module called cross modality reconstruction (CMR-Net) in the multimodal feature fusion stage.

2. A plug-and-play cross modality reconstruction (CMR) module is designed to fuse multi-modal RS data features in a more concise manner to facilitate efficient information exchange and feature integration.

The main significance of this paper is to design a new classification method for multi-modal RS image classification. Beyond the pixel-oriented feature extraction and fusion scheme, a multi-modal RS data classification network based on CNN feature extraction and cross-modal fusion reconstruction is realized. That is, a new cross-modal fusion reconstruction (CMR) module was introduced to reconstruct the features of different modalities in a cross-channel manner, so as to obtain more compact feature representations from RS data of different modalities and realize effective information transmission. Finally, the superiority of the proposed method is verified by experiments, and the classification of multi-modal RS images is realized.

The structure of this paper is as follows: Section II (Related work) provides the related work describes the research work related to multimodal RS classification. Section III (Materials and methods) provides a comprehensive account of the network architecture and associated theories underpinning CMR-Net, encompassing the CNN-based multichannel RS image data classification framework and feature fusion method based on multimodal cross-fertilization reconstruction. In Section IV (Results and discussion), we conducted experiments using HS-LiDAR Houston2013 data and HS-SAR Berlin data, which were subsequently compared with state-of-the-art multichannel classification methods. Our findings are presented in Section V (Conclusions), along with a discussion of potential avenues for future research.

## Related work

There are two main categories of methods for feature extraction and classification of multi-modal RS data, namely morphological profiles (MPs) and subspace learning. Liao et al. [13] fused HSI and LiDAR data on a non-Euclidean space (i.e., manifold) through a graph-based subspace embedding approach. However, this method is prone to lose feature information and its computational cost is high. Ghamisi et al. [14] proposed a joint extraction of attribute profiles (APs) instead of MPs from hyperspectral (HS) and LiDAR data for land cover classification. However, the method is difficult to handle complex scenes, such as intersecting features and shadows, which may lead to inaccurate classification results. Xia et al. [15] developed an integrated classifier for classification tasks using HSI and LiDAR. However, this method has a high computational overhead despite the improvement in classification accuracy.

Among the approaches to subspace learning, Camps-Valls et al. [16] proposed a generalized kernel framework for classification and change detection of multi-temporal and multi-source RS data. Yan et al. [17] assessed the similarity of multi-channel RS data using both Euclidean distance-based and angular distance-based embeddings. However, there are noisy images, leading to inaccurate classification results. Hong et al. [18–20] performed learning and regression via a common subspace on HSI and MS multimodal data. This shared subspace can better convey and interact information from different modalities and is suitable for multimodal and cross-modal RS data classification tasks. Meanwhile, the researchers investigated different regularization terms, such as L1 regularization term and L2 regularization term, and concluded that L1-paradigm regression is more advantageous in multimodal RS data classification. In addition, Hong et al. continued to extend the research work by learning to align the stream structure between two modalities for land cover and land use classification. Hu et al. [21,22] studied the theory of topological analysis in depth and designed a new mapper-based semi-supervised fusion method, which is an extension of the stream alignment method in the literature [23], which is more suitable for HSI and very SAR data classification. However, these methods use shallow feature classification models, which make it difficult to handle complex sample data and nonlinearities. In addition, since these methods rely heavily on a priori information, there are challenges in improving classification accuracy.

With the use and advantages of deep learning networks in RS image classification, researchers have proposed various multimodal RS data classification methods. These methods have been utilized for RS image classification, surpassing the performance bottleneck of using single channels and achieving better results. The multichannel data fusion strategy plays a pivotal role in determining the performance of the multichannel network in classification tasks within RS data. It is important to note that the key factor in the performance of multimodal networks is the fusion strategy, which can be categorized into two types, i.e., cascade-based fusion and alignment-based fusion. Typically, cascade fusion is done by directly superimposing features at early, mid, or late stages, i.e., early fusion, mid fusion, and late fusion, while aligned fusion, i.e., alignment of different modalities by similarity measures or constraints, is effectively realized. In order to optimize the feature fusion stage of HSI and LiDAR data, hang et al. [24] proposed a coupled CNN network and designed both feature-level fusion and decision-level fusion strategies to further improve classification performance. Hong et al. [25] developed a simple and effective encoder-decoder network, called EndNet, for classification of HSI and LiDAR data. Gadiraju et al. [26] proposed a multimodal deep learning framework for crop classification by integrating multispectral and multitemporal satellite images, demonstrating its effectiveness in achieving high accuracy. Suel et al [27] employed multimodal deep learning techniques to estimate the national income of a city, evaluate street congestion, and predict environmental damage by utilizing RS data and street view images. Zhang et al. [28] proposed

a new cross-aware CNN to fuse heterogeneous information and improve joint classification accuracy. Although the above deep learning methods on CNNs significantly improve the classification performance of HSI and LiDAR data, however, the limited training data and feature redundancy lead to a relatively high computational cost. Roy et al. [29] proposed a multimodal fusion transformer for joint classification of HSI and LiDAR. The method uses LiDAR data as learnable tokens for feature learning along with HSI tokens. This operation does not adequately fuse the valid information from both data, thus limiting the accuracy of the classification. In order to fully utilize the information in HSI and LiDAR data to improve the classification accuracy, a new multimodal fusion strategy is designed in this paper to improve the classification accuracy. Guo et al. [30] proposed an unsupervised cross-domain feature fusion and supervised classification network (UF2SCN) for coastal wetland classification. The model initially devises an unsupervised single-branch end-to-end network to acquire the fused features of HSI and LiDAR data, employing a spectral attention feature extraction model to capture the average distribution features of all samples while utilizing HSI and LiDAR data as guiding factors throughout the process. Subsequently, a supervised classification network incorporating spatial attention is employed to classify the fused features using a limited number of samples. Finally, a two-stage training strategy is introduced to enhance the capability of feature fusion. To address the challenge of incomplete information in single-source RS images for ground observation, Feng et al [31] proposed a spectral-spatial-elevation fusion transformer (S2EFT) network for classification. This study introduces the transformer framework into multi-source remote sensing image classification and incorporates two simple yet effective modules: spatial information recognition module and sliding group spectral embedding module. Additionally, traditional CNN commonly used patch form is employed as input data. These improvements effectively tackle the limitations of transformers in focusing on local information, reducing redundant spatial details, and enhancing the transferability of multidimensional pixel information at each location. To address the challenge of using a pre-trained base model directly for multimodal remote sensing data classification due to significant differences between training datasets, He et al [32] proposed a Base Model Adaptive (FMA) framework that does not require parameter fine-tuning. This framework incorporates two learnable modules, namely the cross-space interaction module and the cross-channel interaction module, to extract modality-specific representations from the base model. The cross-spatial and cross-channel interaction modules capture single-peak features from spatial and channel dimensions respectively. Additionally, an FMA-based alignment method (FMA2) is introduced to effectively handle inter-modality differences by establishing a coupling score function, thereby enhancing classification performance.

## Materials and methods

The network structure of the proposed CMR-Net multi-channel RS image data classification is illustrated in Fig 1 in this paper. the CMR-Net network consists of two sub-networks, which are composed of a CNN-based feature extraction sub-network and a fusion sub-network CMR composed of a feature cross-fusion reconstruction-based feature. Fig 2 shows the CNN-based feature extraction process.

### Feature extraction (CNNs)

CNNs extract hierarchical feature representations from diverse data sources, facilitating effective information fusion, particularly for heterogeneous data (e.g., from disparate sensors), which are often challenging to integrate optimally. Suppose $X_1 \in R^{d_1 \times N}$, $X_2 \in R^{d_2 \times N}$ denote different RS data sources, $d_1$, $d_2$ denote dimensions, $N$ are pixels, $x_{1,i} \in X_1$ denotes the ith pixel

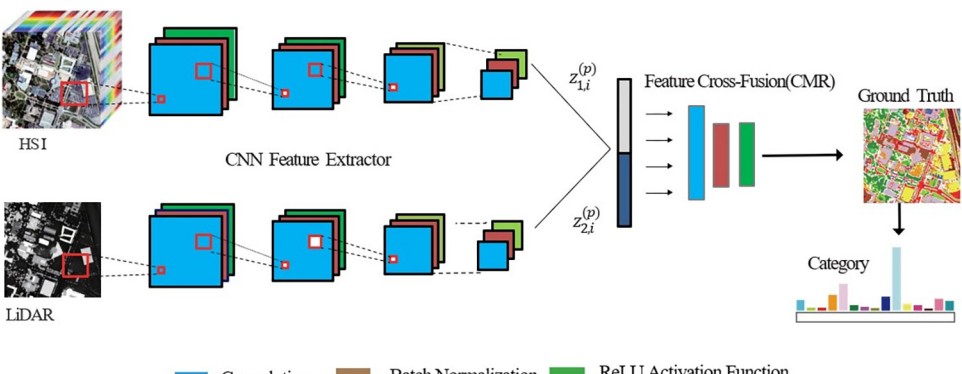

**Fig 1. Illustrates the network architecture of CMR-Net designed for classifying multi-channel RS image data.**
(Charts are similar but not identical to the original images and are therefore for reference only).

pair of the 1st modality, $x_{2i} \in X_2$ denotes the ith pixel pair of the second modality, and these two forms of data sources share the same labeled data. The label matrix is a one-hot coded representation, denotes $Y \in R^{C \times N}$, where C denotes the number of categories and N denotes the number of pixels. The $l$th layer output is represented as

$$z_{s,i}^{(l)} = \begin{cases} h_{W_s^{(l)}, b_s^{(l)}}(x_{s,i}), l = 1 \\ h_{W_s^{(l)}, b_s^{(l)}}(z_{s,i}^{(l-1)}), l = 2, \cdots, p \end{cases} \quad (1)$$

The variable $s = 0,1,2$ represents distinct network streams in this context. in fact, $s = 1,2$ denotes diverse data sources, $p$ denotes the number of convolutional layers, the fused data stream is denoted as $s = 0$, while the linear regression function is represented by $h(\cdot)$, i.e., $h(\cdot) = h(W_s^{(l)} x_{s,i} + b_s^{(l)})$, $\{W_s^{(l)}\}_{l=1}^{P}$ is the learning weight, and $\{b_s^{(l)}\}_{l=1}^{P}$ is the offset. The batch normalization (BN) operation is introduced to accelerate network convergence, mitigate the issue of vanishing or exploding gradients through self-covariate shift among samples, and apply batch regularization to output layer $z_{s,i}^{(l)}$.

$$z_{BN_{s,i}^{(l)}} = \gamma_s \hat{z}_{s,i}^{(l)} + \beta_s \quad (2)$$

where, $\hat{z}_{s,i}^{(l)}$ represents the outcome of z-score normalization applied to the $z_{s,i}^{(l)}$ data, while $\gamma_s$ and $\beta_s$ denote the learning parameters of the network. Before proceeding to the next module, a nonlinear activation function is applied to $z_{BN_{s,i}^{(l)}}$ and the output result $a_{s,i}^{(l)}$.

$$a_{s,i}^{(l)} = u(z_{BN_{s,i}^{(l)}}) \quad (3)$$

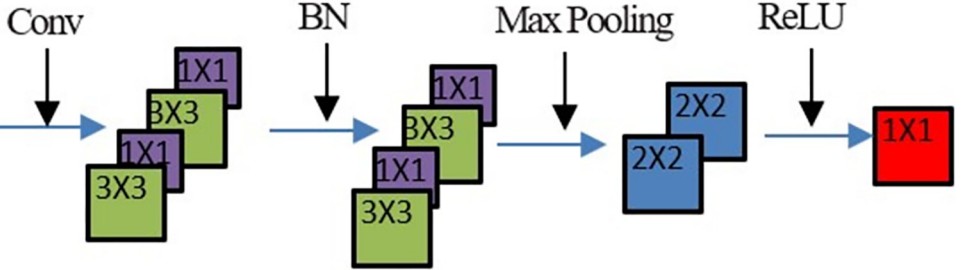

**Fig 2. Feature extraction process of the proposed CMR-net.**

where, $u(\cdot)$ represents the ReLU nonlinear activation function

$$u(\cdot) = \max(0, \cdot) \tag{4}$$

In the feature extraction phase, each modal flow consists of four convolutional blocks, the 1st convolutional block consists of a 3×3 convolutional layer, a BN layer, and a ReLU activation layer, the 2nd convolutional block consists of a 1×1 convolutional layer, a BN layer, a ReLU activation layer, and a 2×2 maximum pooling layer, the 3rd convolutional block consists of a 3×3 convolutional layer, a BN layer, and a ReLU activation layer, and the 4th convolutional block consists of a 1×1 convolutional layer, a BN layer, a ReLU activation layer, and a 2×2 maximum pooling layer.

## Feature fusion (CMR)

Previous multimodal feature fusion methods have generally followed cascade-based fusion strategies, such as early fusion, intermediate fusion, and late fusion. Despite the success of cascade-based fusion methods in feature extraction and representation, their ability to fuse different attributes, especially for heterogeneous data, is still limited. Efficiently integrating features from diverse modalities is a highly effective approach. In this paper, we propose a plug-and-play fusion module, i.e., cross modality reconstruction fusion (CMR), where the CMR module learns more compact features among different modalities by constantly updating the parameters of different sub-networks, and based on such a network setup, not only can it learn its own features, but also learns more diversified features from another channel of the network flow to learn more diverse features as an effective information fusion and feature-level fusion method. Fig 3 shows the process of the CMR feature reconstruction module. After extracting the coded features from the input multichannel RS data $X_1$ and $X_2$ by feature extraction network, it is noted as $\{A_s = [a_{s,1}^{(p)}, \cdots, a_{s,N}^{(p)}]\}_{s=1}^{2}$, as a new input, which is provided to the CMR-Net network for feature fusion, $A_s$ denotes the set of features after feature extraction, $s$ denotes the number of data sources, i.e. the number of multimodalities. $p$ denotes the number of convolutional blocks, N denotes the number of pixels, the output of the fused CMR-Net network is:

$$a_i^{(l)} = f_{W_s^{(l)}, b_s^{(l)}}(a_{s,i}^{(p)}), l = p+1, \cdots, q \tag{5}$$

where, the function $f(\cdot)$ denotes a nonlinear mapping function consisting of several blocks in the fusion network. $a_i^{(l)}$ denotes the ith pixel after feature fusion, $l$ denotes the lth convolutional block. The CMR-Net network aims to learn better features on multiple modalities. Taking the

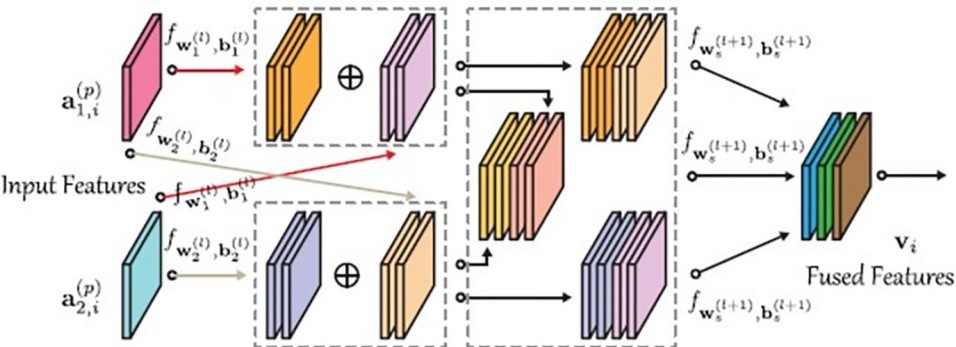

**Fig 3. Shows the procedure of the CMR module in the proposed CMR-net.**

$i$th pixel as an example, the fusion is expressed as

$$a_{1,i}^{(l)} = f_{W_1^{(l)}, b_1^{(l)}}\left(a_{1,i}^{(p)}\right) + f_{W_1^{(l)}, b_1^{(l)}}\left(a_{2,i}^{(p)}\right) \tag{6}$$

$$a_{2,i}^{(l)} = f_{W_2^{(l)}, b_2^{(l)}}\left(a_{2,i}^{(p)}\right) + f_{W_2^{(l)}, b_2^{(l)}}\left(a_{1,i}^{(p)}\right) \tag{7}$$

$$v_i = \begin{bmatrix} f_{W_0^{(l+1)}, b_0^{(l+1)}}\left(a_{1,i}^{(l)}\right), & f_{W_0^{(l+1)}, b_0^{(l+1)}}\left(a_{2,i}^{(l)}\right) \\ f_{W_1^{(l)}, b_1^{(l)}}\left(a_{1,i}^{(p)}\right), & f_{W_2^{(l)}, b_2^{(l)}}\left(a_{1,i}^{(p)}\right) \\ f_{W_1^{(l)}, b_1^{(l)}}\left(a_{2,i}^{(p)}\right) & f_{W_2^{(l)}, b_2^{(l)}}\left(a_{2,i}^{(p)}\right) \end{bmatrix} \tag{8}$$

where, the three components of $v_i$ in (8), i.e., each row of the $v_i$ matrix, share the same parameters to be learned. $a_{1,i}^{(l)}$ denotes the $i$th pixel fusion of the first modality, $a_{2,i}^{(l)}$ denotes the $i$th pixel fusion of the second modality, and $v_i$ is the matrix after feature fusion. Fig 3 illustrates the interaction process in feature fusion in CMR-Net network, where a highly compact fusion is achieved by cross weighting and features to achieve "better" and "more efficient" fusion representation. More specifically, the learned weights can be used across modalities, e.g., the weights learned from modality A can be applied to modality B at the same time, and vice versa. The features after the summation operation are then output and the features are cross-combined again as the final fused representation of the cross-fusion.

## Function optimizes

The parameter tuning of the CMR-Net network proposed in this paper is iteratively updated by the following loss function: here, $\beta$ is the penalization factor of

$$L = L_l + \beta L_{loss} \tag{9}$$

In this paper the parameter $\beta$ is determined based on empirical values and is given a value of 1 in the paper to minimize the volatility of its loss function. The function $L_1$ represents the cross-entropy loss between the fusion feature $v_i$ and one-hot label $y_i$, $N$ are pixels, which can be expressed as follows:

$$L_l = \frac{-1}{N} \sum_{i=1}^{N} [y_i \log v_i + (1 - y_i) \log(1 - v_i)] \tag{10}$$

and $L_{loss}$ is the loss of CMR-Net, which is expressed as

$$L_{loss} = \sum_{i=1}^{N} \left| [a_{2,i}^{(p)}, a_{1,i}^{(p)}] - v_i \right|_2^2 \tag{11}$$

where $v_i$ denotes the output feature vector after reconstruction, and $L_{loss}$ calculates the loss of the reconstructed fused feature $v_i$ and multimodal feature $[a_{2,i}^{(p)}, a_{1,i}^{(p)}]$ after $L_2$ paradigm regularization.

## Results and discussion

### Data description

**Houston 2013 data.** The ITRES CASI-1500 image sensor was utilized for the acquisition of this dataset. The research facility is situated within the University of Houston campus and

**Table 1. The number of training and test sets for the Houston2013 dataset will be described.**

| Class No | Class Name | Training Set | Testing Set |
|---|---|---|---|
| C1 | Healthy Grass | 198 | 1053 |
| C2 | Stressed Grass | 190 | 1064 |
| C3 | Synthetic Grass | 192 | 505 |
| C4 | Tree | 188 | 1056 |
| C5 | Soil | 186 | 1056 |
| C6 | Water | 182 | 143 |
| C7 | Residential | 196 | 1072 |
| C8 | Commercial | 191 | 1053 |
| C9 | Road | 193 | 1059 |
| C10 | Highway | 191 | 1036 |
| C11 | Railway | 181 | 1054 |
| C12 | Parking Lot1 | 192 | 1041 |
| C13 | Parking Lot2 | 184 | 285 |
| C14 | Tennis Court | 181 | 247 |
| C15 | Running Track | 187 | 473 |
| | Total | 2832 | 12197 |

its surrounding rural environs in Texas, USA. This dataset comprises two data sources, one of which is Hyperspectral (HS) imagery that encompasses 144 spectral channels ranging from 364 to 1046 nm with a spectral resolution of 10 nm. The alternative data source comprises a LiDAR image with dimensions of 349 × 1905 pixels, wherein a single band is utilized to provide height information for the corresponding image region. The studied scenarios encompass a total of 15 land cover and land use categories. Table 1 presents the names of these categories included in the scenarios, along with the respective sizes of the training and test sets.

**Berlin data.** This dataset provides comprehensive feature data pertaining to the urban and rural areas of Berlin. The dataset comprises two distinct data sources: a HS image encompassing 797 × 200 pixels, 244 spectral channels, and wavelengths spanning from 400 to 2500 nm; and a polarized SAR image consisting of four bands, namely VV-VH. In addition, the real-world scene maps are derived from free open street maps. Table 2 presents the spectral information, which is categorized into training and test sets along with their respective sizes for both sets.

## Experimental setup

**Evaluation metric.** In this paper's experiments, the classification performance of multi-modal RS data is evaluated using three commonly used metrics: Overall Accuracy (OA),

**Table 2. The number of training and test sets for the Berlin dataset will be described.**

| Class No | Class Name | Training Set | Testing Set |
|---|---|---|---|
| C1 | Forest | 443 | 54511 |
| C2 | Residential Area | 423 | 268219 |
| C3 | Industrial Area | 499 | 19067 |
| C4 | Low Plants | 376 | 58906 |
| C5 | Soil | 331 | 17095 |
| C6 | Allotment | 280 | 13025 |
| C7 | Commercial Area | 298 | 24526 |
| C8 | Water | 170 | 6502 |
| | Total | 2820 | 461851 |

Average Accuracy (AA), and kappa coefficient ($\kappa$). OA, AA, and $\kappa$ are commonly used evaluation metrics for evaluating the performance of classification models, which are also suitable for classification models of convolutional neural networks (CNN). These metrics provide a test of accuracy and agreement between the predicted and true labels. OA measures the percentage of correctly classified samples over the total number of samples in the dataset. It provides a generic measure of model accuracy. AA measures the average precision of each class. It provides an indicator of the classification performance of the model in different categories. The $\kappa$ measures the agreement between the predicted and true labels, considering agreement that may occur by chance. It is a more robust metric than OA, especially when dealing with imbalanced datasets. The $\kappa$ coefficient ranges from [-1 to 1], where closer to 1 indicates stronger agreement. These evaluation metrics provide different perspectives on the performance of the model. OA gives an overall accuracy measure, AA evaluates the accuracy of each category, and $\kappa$ measures consistency. They are often used together to provide a comprehensive assessment of the performance of the classification model. Generally, higher values for these metrics indicate better classifier performance in RS image classification tasks. For those models based on feature learning, we first learn multimodal feature representations, then apply classifiers, and finally perform quantitative performance comparisons with other methods. More specifically, OA, AA, and $\kappa$ provide quantitative evaluations of classification performance, with their formulas as follows.

$$OA = \frac{N_c}{N_a} \tag{12}$$

$$AA = \frac{1}{C} \sum_{i=1}^{C} \frac{N_c^i}{N_a^i} \tag{13}$$

and

$$\kappa = \frac{OA - P_e}{1 - P_e} \tag{14}$$

where, $N_c$ denotes the number of correctly classified samples, $N_a$ denotes the number of all samples, $N_c^i$ denotes the number of correctly classified samples in the ith category among the number of correctly classified samples, and $N_a^i$ denotes the number of correctly classified samples in the i-th category among all samples. In $\kappa$, $P_e$ is defined as the expected prior probability and is calculated as

$$P_e = \frac{N_r^1 \times N_p^1 + \dots N_r^i \times N_p^i + \dots + N_r^C \times N_p^C}{N_a \times N_a} \tag{15}$$

where $N_r^i$ represents the count of true samples in each category while $N_p^i$ denotes the count of predicted samples in each category.

## Implementation details

The CMR-Net network experiments proposed in this paper run on the TensorFlow platform implementation, running on a host environment with an Intel Xeon processor and NVIDIA GeForce GTX 1080Ti's GPU processor. The network model was trained on the training set, and the hyperparameters were determined via a grid search performed on the validation set. Specifically, 10 repetitions were conducted to randomly partition the original training set into new training and validation sets with an 8:2 ratio for determining hyperparameters of the final

network. During the training phase, we employ the Adam optimizer with an "exponential" learning rate that can be updated by multiplying the baseline learning rate by (1−(inter/maxIter)) raised to a power every 30 cycles. The learning rate and exponential power (power) are initialized to 0.001 and 0.5, respectively. Due to the random nature of the initialization, the initialized learning rate is set to 0.001 every 10 executions. Randomness, the average result is displayed every 10 executions; the network training initialization sets the momentums variable to 0.9; the batch training block is set to 64. In order to facilitate the network training as well as avoid overfitting, regularization is employed to adjust the weights in order to prevent overfitting, thereby enhancing the generalizability of the model, and the network stops training when the loss rate of the validation set can not be reduced. The CMR-Net network consists of two parts, CNN feature extraction and feature fusion. The input convolution kernel size is set to 0.001 and 0.5, respectively. In parts, the input convolutional kernel size is set to 7×7-pixel size, and the CNN feature extraction module includes two modalities: 4 convolutional layers, 4 BN layers, 2 maximal pulling layers and 4 ReLU layers. The CMR feature fusion module includes 9 CNN convolutional layers, 6 BN layers, 3 average pooling layers and 6 ReLU layers. To expedite the rate of convergence and classification accuracy of the training parameters, the epoch is set to 150 according to experience, which can better achieve the convergence of the model.

Special attention is paid to the fact that in the process of experimentation, an appropriate increase in the number of layers of convolutional layers can improve the performance and accuracy of classification, but it will increase the cost of computation, and the network convergence speed will become slower. The CMR-Net network proposed in this paper is mainly to achieve the advantage of feature representation for multimodal RS images, without tuning and setting the network parameters. In order to compare fairly with other network structures, the CMR-Net network structure designed in this paper is basically the same as other network structures, and the network parameters and complexity are also similar.

## Comparison with state-of-the-art multimodal algorithms

In order to demonstrate the effectiveness of the CMR-Net method proposed in this paper, several state-of-the-art methods for multimodal classification are selected for quantitative and qualitative comparison. Specifically, e.g., HS, LiDAR with single modal features, HSI-LiDAR and HSI-SAR with multimodal superimposed features, CapsNet [33], LeMA [22], Co-CNN [24], EndNet [25] and our CMR-Net.

## Result and analysis on houston data

**Quantitative comparison.** Table 3 presents the quantitative classification results of various multichannel methods based on three commonly used metrics, namely OA, AA and κ. Fig 6 visualizes the corresponding classification maps obtained by these methods.

In general, the classification results of HS data are significantly higher than those of LiDAR data using RF classifier at the same time, while better classification performance than single-modal HS and LiDAR data is obtained by superimposing multimodal data with HS and LiDAR features, i.e., RF (H+L), such as the Random Forest (RF) classification method, as shown in Table 3. This experiment demonstrates that using multimodal RS data in feature extraction and classification is better than using single modal data for classification and more effective classification methods. In traditional classification using semi-supervised methods, LeMA fully considers labeled and unlabeled samples to find better decision boundaries and achieve higher classification accuracy. Among them, category C3 and category C14 are categorized with complete accuracy. Unlike traditional subspace-based and morphology-based approaches, deep learning-based models, such as CapsNet, Co-CNN, EndNet, and our

**Table 3. Quantitative comparison of various methods in terms of overall accuracy (OA), average accuracy (AA), and kappa coefficient (κ) on the HS-LiDAR Houston 2013 datasets.**

| Methods | RF-HS | RF-LiDAR | RF(H+L) | CapsNet | LeMA | Co-CNN | EndNet | CMR-Net |
|---|---|---|---|---|---|---|---|---|
| C1 | 82.62 | 13.49 | 82.53 | 81.10 | 81.86 | 83.10 | 96.13 | 99.30 |
| C2 | 83.55 | 16.26 | 83.36 | 81.02 | 83.08 | 84.87 | 94.72 | 90.52 |
| C3 | 97.62 | 56.63 | 98.42 | 96.44 | 100.00 | 99.80 | 98.73 | 94.80 |
| C4 | 91.95 | 42.80 | 92.81 | 88.35 | 94.79 | 92.42 | 91.50 | 96.54 |
| C5 | 96.78 | 58.05 | 96.63 | 100.00 | 99.34 | 99.24 | 98.98 | 98.86 |
| C6 | 99.30 | 58.04 | 99.60 | 95.80 | 99.30 | 95.80 | 93.22 | 60.71 |
| C7 | 74.72 | 41.42 | 80.04 | 86.38 | 88.99 | 95.24 | 89.25 | 84.82 |
| C8 | 33.43 | 27.64 | 43.02 | 90.88 | 74.26 | 81.86 | 76.52 | 89.75 |
| C9 | 70.16 | 14.35 | 69.97 | 82.53 | 73.84 | 85.08 | 86.05 | 93.46 |
| C10 | 43.24 | 9.07 | 45.85 | 72.78 | 72.20 | 61.10 | 91.28 | 93.80 |
| C11 | 69.83 | 39.94 | 74.31 | 88.99 | 82.26 | 83.87 | 87.60 | 92.09 |
| C12 | 54.85 | 8.55 | 67.12 | 83.09 | 90.30 | 91.26 | 83.18 | 87.70 |
| C13 | 60.00 | 15.44 | 64.56 | 76.14 | 67.37 | 88.77 | 84.18 | 92.99 |
| C14 | 99.60 | 80.16 | 99.97 | 93.93 | 100.00 | 91.09 | 98.33 | 98.02 |
| C15 | 97.89 | 75.90 | 97.67 | 97.46 | 98.10 | 98.73 | 98.99 | 95.51 |
| OA (%) | 73.12 | 31.49 | 78.79 | 86.52 | 85.42 | 87.23 | 90.18 | 92.12 |
| AA (%) | 7.74 | 37.18 | 79.20 | 87.66 | 87.05 | 88.82 | 90.28 | 91.26 |
| κ | 32.00 | 26.36 | 77.15 | 85.41 | 84.17 | 86.19 | 88.72 | 91.50 |

CMR-Net in this paper, are able to extract more discriminative features and better capture potential relationships between different modalities. This enables the three deep learning models, CapsNet, Co-CNN and EndNet, to obtain better classification results. Although capsule units are applied in CapsNet to extract richer features, the ability of feature fusion is still limited, especially for heterogeneous RS data. This leads to some extent to a slightly worse performance of CapsNet than Co-CNN. Although Co-CNN is better than CapsNet, Co-CNN fuses features based on a double-coupled cascade with noise, which results in Co-CNN classification accuracy lower than that of EndNet by about 3%. It should be noted that the CMR-Net method

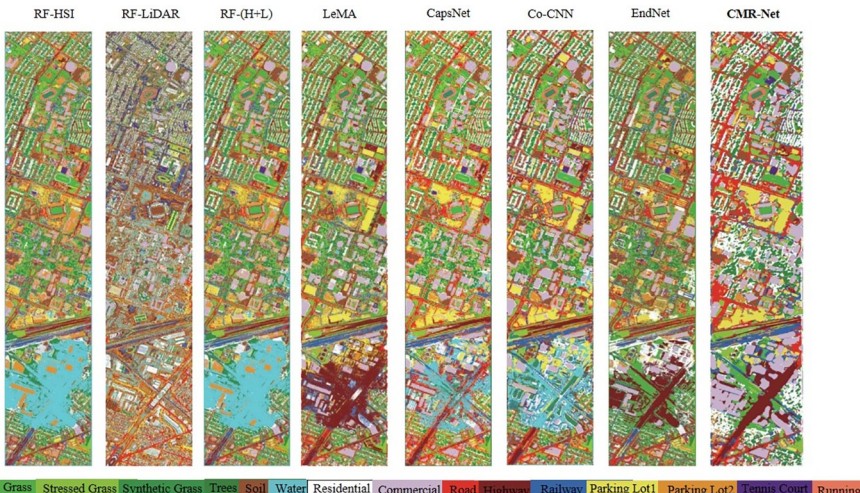

**Fig 4. Classification maps of different multi-modality methods on the HS-LiDAR Houston2013 data.** (Graphs are similar but not identical to the original images, comparing image sources with literature [18,21,22]).

proposed in this paper overcomes the inadequacy of other methods in feature fusion, reduces duplicated features, obtains more discriminative features, and thus outperforms other methods in multimodal data fusion and classification. Better classification results are achieved.

## Visual comparison

The classification effect graph of the proposed CMR-Net method with other methods is shown in Fig 4. In this paper, in addition to quantitative analysis, we also validate the effectiveness of the proposed CMR-Net method from a visual perspective. In addition, compared with other state-of-the-art methods, the classification maps generated by our CMR-Net can provide more realistic land cover and land use markers and provide better spatial aggregation results. CMR-Net can eliminate the effect of cloud cover from the visual point of view to a certain extent. In addition, our CMR-Net tends to generate more realistic and smoother classification maps with fewer noisy pixels (misclassified points), while discovering important visual, spectral, and other information from highly complex images.

## Performance analysis

Fig 5 shows the convergence and stability of the CMR-Net network proposed in this paper under the training and test samples of the Houston 2013 dataset. As depicted in Fig 5, it is evident that the number of iterations of the samples has a direct impact on the results, the loss rate of the training and test sets gradually decreases, gradually converges, the convergence rate gradually tends to be flat until it is completely converged, the loss rate tends to be stable, and the model tends to be stable. It can be seen that the training samples and test samples in the model training, the number of iterations by half, the loss rate tends to be stable, and the model is basically stable. Conversely, as the number of model iterations increases, the model tends to exhibit stability when trained 50 times. The classification accuracy of the training set data remains consistently high and unchanged, while the classification accuracy of the test set data gradually converges. And when the number of iterations is about 100, the classification accuracy of the test set data fluctuates very little and stays stable, and the accuracy of classification reaches about 92%. This proves to a large extent the high stability and fast convergence of the CMR-Net network proposed in this paper.

## Result and analysis on Berlin data

**Quantitative comparison.** Similar to the quantitative analysis of the HS-LiDAR Houston2013 dataset, the quantitative classification performance of the HS-SAR Berlin dataset is

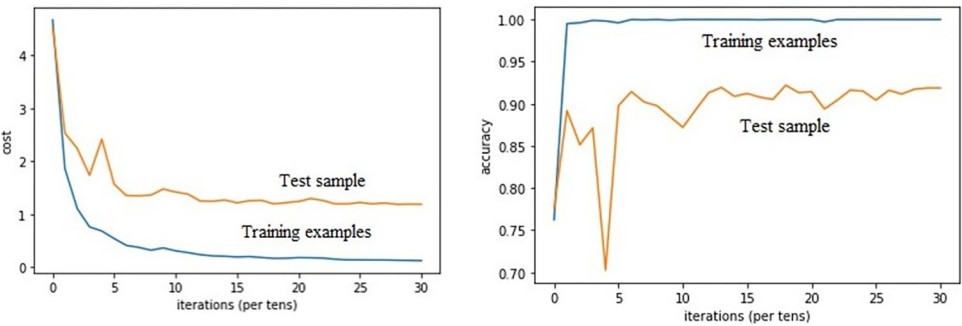

**Fig 5. The classification performance of the proposed CMR-NET network is evaluated on HS-LiDAR Houston2013 dataset.**

**Table 4. Quantitative comparison of various methods based on OA, AA, and κ using the HS-SAR Berlin datasets.**

| Methods | RF-HS | RF-SAR | RF(H+L) | CapsNet | LeMA | Co-CNN | EndNet | CMR-Net |
|---------|-------|--------|---------|---------|------|--------|--------|---------|
| C1 | 60.29 | 31.52 | 78.35 | 84.86 | 84.11 | 83.09 | 88.25 | 89.18 |
| C2 | 58.78 | 29.72 | 60.46 | 65.22 | 64.84 | 68.48 | 67.36 | 68.07 |
| C3 | 40.74 | 35.60 | 42.64 | 48.32 | 42.53 | 48.09 | 54.26 | 55.17 |
| C4 | 62.58 | 43.07 | 65.02 | 80.70 | 80.04 | 78.43 | 83.22 | 83.62 |
| C5 | 66.15 | 51.00 | 67.84 | 69.08 | 80.66 | 80.25 | 86.02 | 86.10 |
| C6 | 60.19 | 32.37 | 61.01 | 55.08 | 54.07 | 48.70 | 63.69 | 64.02 |
| C7 | 25.01 | 12.04 | 34.85 | 26.11 | 27.40 | 25.16 | 31.25 | 32.23 |
| C8 | 66.90 | 33.88 | 67.88 | 59.59 | 57.75 | 58.52 | 72.64 | 72.78 |
| OA (%) | 52.02 | 32.36 | 61.45 | 67.05 | 66.21 | 67.45 | 70.05 | 71.95 |
| AA (%) | 55.08 | 33.65 | 59.75 | 61.12 | 62.05 | 61.34 | 68.34 | 68.78 |
| κ | 40.34 | 17.10 | 50.02 | 52.07 | 52.12 | 54.36 | 59.71 | 60.16 |

shown in Table 4. The analysis of classification performance and the trend of the number of sample iterations and computational overhead are basically the same as the HS-LiDAR Houston2013 dataset. The classification results in Table 4 show that the fusion of HS and SAR data sources can effectively improve the classification accuracy. Comparing the classification effect from a single data source, HS data has stronger feature discrimination than SAR data. In traditional classification using semi-supervised methods, Features learned through joint feature learning models, such as LeMA, are more inclined to find better decision boundaries for classifiers, and achieve higher classification accuracy. Unlike traditional subspace-based and morphology-based approaches, deep learning-based models, such as CapsNet, Co-CNN, EndNet, and our CMR-Net in this paper, are able to extract more discriminative features and better capture potential relationships between different modalities. Although capsule units are applied in CapsNet to extract richer features, the ability of feature fusion is still limited, especially for heterogeneous RS data. C3 and C7 feature classes in the SAR data are similar, the fusion features are noisy, and the difference in classification accuracy between CapsNet and Co-CNN is small. EndNet method can reduce noise to a certain extent by encoding-decoding fusion of fused features and obtain better discriminative features, which improves classification accuracy by about 3% compared with Co-CNN. It should be noted that the CMR-Net method proposed in this paper overcomes the inadequacy of other methods in feature fusion, reduces duplicated features, obtains more discriminative features, and thus outperforms other methods in multimodal data fusion and classification. Better classification results are achieved.

**Visual comparison.** The classification performance diagrams of the CMR-Net architecture proposed in this study and other multimodal RS data classification networks are depicted in Fig 6. Through intuitive observation, it can be visually seen that the classification maps generated based on deep learning methods, especially our proposed CMR-Net, use an effective fusion of spatial and spectral information, and thus classify the effect maps more smoothly than the other methods in this paper. Similarly, CMR-Net tends to produce better visual fidelity, especially for classifying residential and forested areas, and can recognize feature information more efficiently and accurately. Additionally, the CMR-Net classification approach is capable of extracting crucial visual, spectral, and other information from intricate scenes, thereby generating precise classification outcomes that closely reflect reality.

## Result and discussion

In this paper, the proposed CNN-based classification method for multimodal RS image data with cross-modal reconstruction compactly integrates the multimodal RS features for effective

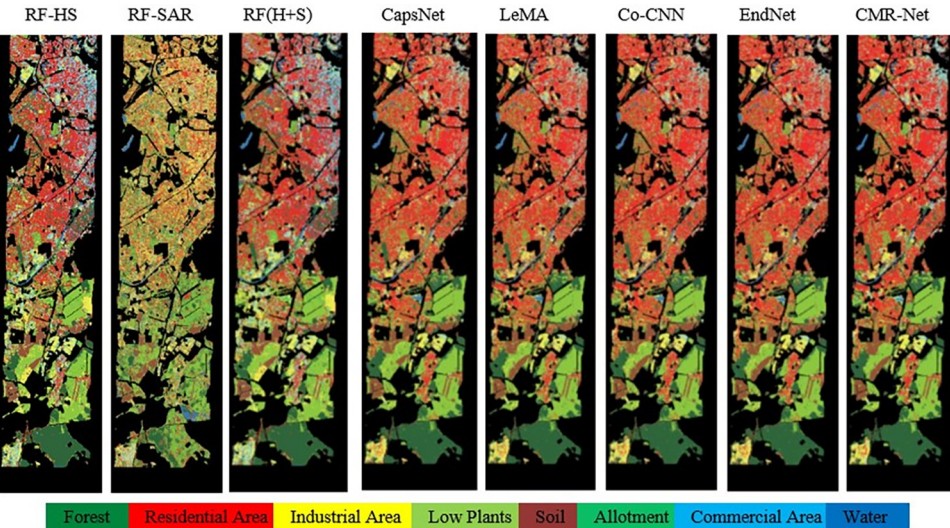

**Fig 6. Presents the classification maps of CMR-Net applied to the HS-SAR data.**

information exchange and feature fusion. The results show that our proposed CMR-Net's is extremely competitive compared to several state-of-the-art multimodal RS classification methods. The CMR-Net method proposed in this paper is based on being inspired by the methods proposed by previous researchers and summarizing the advantages and disadvantages of the previously proposed methods. The Co-CNN method proposed in the literature [24] fuses HS and LiDAR data using a learning coupling and cascading approach, but it only implements a simple superposition computation of the features, with less feature overlap and noise consideration. Later, in order to better improve the classification accuracy and reduce the noise, in the literature [25], Hong et al. designed a simple and effective encoder-decoder network, called EndNet, for the classification of HS and LiDAR data, and even though the feature overlap and noise problems are solved to a certain extent, and the classification accuracy is improved, the features need to be encoded and decoded, and the computational overhead is increased. The CMR-Net network proposed in this paper is based on the methods proposed in the literature [24,25], and introduces the CMR module, which realizes the effective fusion of multimodal features through cross-modal feature reconstruction, reduces the feature noise, and obtains effective feature identification by full information interaction between the features of different data sources, which improves the classification effect and classification accuracy.

Although the CMR-Net multimodal classification method proposed in this paper has advantages compared to other classification methods and its classification accuracy is higher, the CMR-Net classification method increases the number of layers of the CNN convolutional layers and spends more computational overhead in order to extract the multimodal features more adequately in the process of implementation, and this method performs cross-modal feature fusion for the extracted features, which is more accurate compared to the other two deep learning methods, it also has to increase the computational cost and the network convergence will be slower. This method of cross-modal feature fusion of extracted features, compared to the other two deep learning methods in the paper, also increases the computational cost, and the speed of network convergence will be slower. By observing during the experiment, the computational overhead of CMR-Net takes about 10 minutes more than EndNet. However, it can be changed by improving the performance of the GPU compute card. In addition, in the

process of experiments and research, the feature fusion of multiple data sources (such as three data sources) is not carried out to verify the effectiveness of the model. Corresponding research will be carried out in the later research.

## Conclusions

In order to better and more efficiently classify multimodal RS data, in this paper, a flexible and plug-and-play CMR module is introduced into the framework of a CNN-based multimodal deep learning network, i.e., the CMR-Net network, which can be used to make the features of different modalities to be fused and reconstructed in a more compact and effective way. The CMR-Net network proposed in this paper comprises two components, namely a CNN-based sub-network for feature extraction and a CMR-based fusion sub-network that utilizes cross-fusion reconstruction of features. The proposed method offers superior advantages, yielding enhanced classification results and improved performance compared to conventional cascade mode fusion and alignment mode fusion techniques. This also proves the superiority and efficiency of the CMR-Net network in multimodal RS data classification work.

This paper also faces some limitations and difficulties in the process of research work, mainly because of the relatively small number of multimodal RS data sources, difficult to obtain, and most of them are small samples of data, which is a limitation for a wider range of research work. At the same time, the CMR-Net method proposed in this paper involves some parameter tuning, which requires a lot of computation time during training and requires a better performance GPU card, which is a limitation for the rapid presentation of computational results.

Although the CMR-Net method proposed in this paper provides new ideas and directions in the work of classifying multimodal land cover, its practicality and application capabilities still need to be further developed, especially in the feature fusion with real scenes and more modal data in complex situations. In order to break through the performance bottleneck of multimodal deep learning, we will introduce weakly supervised or self-supervised techniques into the network of fusion modules in our future research work.

## Acknowledgments

The authors would like to express their gratitude to the Hyperspectral Image Analysis Group at the University of Houston and the IEEE GRSS DFC2013 for generously providing the University of Houston HS dataset. Additionally, special thanks are extended to the Mapping Laboratory of the Department of Geography at Humboldt University, Berlin, for their kind provision of modeled EnMAP hyperspectral data specifically tailored for the urban area in Berlin.

## Author Contributions

**Data curation:** Huiqing Wang, Lingfeng Wu.

**Formal analysis:** Huajun Wang.

**Methodology:** Huajun Wang, Lingfeng Wu.

**Validation:** Lingfeng Wu.

**Writing – original draft:** Huiqing Wang.

**Writing – review & editing:** Huajun Wang.

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
