## [Decision Letter · Decision Letter 0]

9 Oct 2023

PONE-D-23-24147CMR-Net: A Cross Modality Reconstruction Network for Multi-modality Remote Sensing ClassificationPLOS ONE

Dear Dr. Wang,

Thank you for submitting your manuscript to PLOS ONE. After careful consideration, we feel that it has merit but does not fully meet PLOS ONE’s publication criteria as it currently stands. Therefore, we invite you to submit a revised version of the manuscript that addresses the points raised during the review process.

We look forward to receiving your revised manuscript.

Kind regards,

Ashwani Kumar, Ph.D.

Academic Editor

PLOS ONE

Journal Requirements:

"This article research results by the southwest university of medical sciences field scientific research subject (fund no. 20/00031616, 20, 01041224, 20/01041306) support."

5. We note that Figures 1,2 and 4 in your submission contain [map/satellite] images which may be copyrighted. All PLOS content is published under the Creative Commons Attribution License (CC BY 4.0), which means that the manuscript, images, and Supporting Information files will be freely available online, and any third party is permitted to access, download, copy, distribute, and use these materials in any way, even commercially, with proper attribution. For these reasons, we cannot publish previously copyrighted maps or satellite images created using proprietary data, such as Google software (Google Maps, Street View, and Earth). For more information, see our copyright guidelines: http://journals.plos.org/plosone/s/licenses-and-copyright.

a. You may seek permission from the original copyright holder of Figures 1,2 and 4  to publish the content specifically under the CC BY 4.0 license.  

Reviewers' comments:

Reviewer's Responses to Questions

**Comments to the Author**

1. Is the manuscript technically sound, and do the data support the conclusions?

Reviewer #1: Yes

Reviewer #2: Yes

Reviewer #3: Yes

Reviewer #4: Yes

2. Has the statistical analysis been performed appropriately and rigorously? 

Reviewer #1: Yes

Reviewer #2: Yes

Reviewer #3: Yes

Reviewer #4: N/A

3. Have the authors made all data underlying the findings in their manuscript fully available?

Reviewer #1: Yes

Reviewer #2: Yes

Reviewer #3: Yes

Reviewer #4: Yes

4. Is the manuscript presented in an intelligible fashion and written in standard English?

Reviewer #1: Yes

Reviewer #2: Yes

Reviewer #3: Yes

Reviewer #4: Yes

5. Review Comments to the Author

Reviewer #1: -What are the criteria used by the researcher to demonstrate the efficiency of the proposed model? Please indicate this in the abstract

-The researcher did not specify the keywords. Please add

-The research listed as reviewed literature was not arranged according to the year of publication (from oldest to newest). Also, weaknesses were not indicated in each of them, which the researcher might have overcome with his contributions to this research paper.

-Are all the listed equations made by the researcher, or there are equations that were used from a previous research because there was no mention of any reference before any equation was mentioned

-Are there specific links to get the databases? If any, please add them to the references

-The submitted work has not been compared with previous works to demonstrate the efficiency of the system compared to previous works

-Standardization of reference format

Reviewer #2: 1. The paragraphs are not numbered, which should be done so comments and suggestions can be pointed out easily.

2. Chapter arrangements, numbering and formatting should be corrected in line with this journal’s recommended formatting guidelines.

3. The paper does not discuss any potential drawbacks or shortcomings of the CMR-Net method in comparison to other state-of-the-art algorithms. Hence, the authors should provide a detailed discussion on the limitations and challenges faced during the implementation of the CMR-Net architecture, as this would help in understanding the potential drawbacks of the proposed method.

4. Conduct a comparative analysis with a wider range of state-of-the-art algorithms for multi-modality remote sensing data classification, to provide a more comprehensive evaluation of the performance of CMR-Net.

5. Include a discussion on the computational cost and efficiency of the CMR-Net method, as increasing the number of layers in the network may impact the convergence speed and computational requirements.

Reviewer #3: I suggest to add paragraph in last section to explain te most significant improvements they got it from results

I suggest to add table of comparison to compare their work with other

I suggest to add table of state to explain related works with methods , mercies, result, type of CNN and limitation

Reviewer #4: The article is technically sound and the title also justifies the content. However, some changes are required to be made to

for better understanding. Kindly incorporate the suggested changes in the revised manuscript carefully.

6. PLOS authors have the option to publish the peer review history of their article (what does this mean?). If published, this will include your full peer review and any attached files.

Reviewer #1: No

Reviewer #2: No

Reviewer #3: No

Reviewer #4: **Yes: **Purnima

---

## [Author Response · Author response to Decision Letter 0]

20 Oct 2023

Response: I have adjusted the relevant formatting of the article according to the template given by the editorial team. If there is still something wrong, please point it out to the editorial board. Thank you!

Response: I have reviewed the relevant guidance notes as requested to ensure that the code is shared in a way that follows best practice and promotes repeatability and reuse.

"This article research results by the southwest university of medical sciences field scientific research subject (fund no. 20/00031616, 20, 01041224, 20/01041306) support."

 Response: Grantee Lingfeng Wu was involved in the preliminary research design and data collection in the study. Grantee Huajun Wang participated in the preliminary research design and data collection in the study.

 Response: Modify data availability to: No. This is because the experiment was a team effort, and the data involved the work of other members of the team, which could affect their results.

5. We note that Figures 1,2 and 4 in your submission contain [map/satellite] images which may be copyrighted. All PLOS content is published under the Creative Commons Attribution License (CC BY 4.0), which means that the manuscript, images, and Supporting Information files will be freely available online, and any third party is permitted to access, download, copy, distribute, and use these materials in any way, even commercially, with proper attribution. For these reasons, we cannot publish previously copyrighted maps or satellite images created using proprietary data, such as Google software (Google Maps, Street View, and Earth). For more information, see our copyright guidelines: http://journals.plos.org/plosone/s/licenses-and-copyright.

 Response: Thanks for pointing out the picture problems in the article, I have modified the pointed out Fig. 1 and Fig. 2, Fig. 4 is the visualization of the pairs of unanalyzed graphs, the comparative analysis part of the graphs from the literature, CMR-Net and EndNet from the experimental results.

6.The introductory section does not adequately explain the framework and problems of research in the respective area. Elaborate the introduction part and state the motivation clearly.

Response: Thank you very much for the review suggestions, which are very helpful for my later research work. I have followed the suggestions in the "Introduction" and "Related work" sections of the article, respectively, from the traditional methods of multimodal remote sensing classification based on morphological and subspace classification methods, and then to the current deep learning methods in various fields of research frameworks. The research frameworks of various fields from the traditional methods of multimodal remote sensing classification based on morphology and subspace, and then to the current deep learning methods are analysed and explained.

7.Give the rationale of the study that justifies why the specific approach has been used by the author(s).

Response: First of all, I would like to thank you very much for the review suggestions, which are very helpful for my later research work. I have described the background and principle of this research and proposed the specific research methodology in the "Introduction" part of the article according to the suggestions, and then described the current status of the related research in the "Related work" part according to the temporal clues of the development of multimodal remote sensing data categorization.

8.The review of literature section is merged with the introduction part. It should be written under a separate heading ‘Related Work’. Also, the author(s) are advised to add a few latest studies relevant to the area in this section.

Response: First of all, I would like to thank you very much for the review suggestions, which are very helpful for my later research work. According to the suggestion, I have merged the literature review with the introduction, added the section of "Related work", and added the latest research in this field in 2022 in the literature.

9.The clarity of presentation of mathematical foundation of the method is weak. Elaborate the same and the variables used in the equations should be well denoted and explained.

 Response: First of all, I would like to thank you very much for the review suggestions, which are very helpful for my later research work. According to the suggestion, I have clearly revised the mathematical formulas and foundations in the "Materials and methods" part of the article to make them clear, and added detailed descriptions and illustrations of the formulas.

10.The results are not suitably described. It is suggested to put in the details and strengthen the same.

Response: First of all, I would like to thank very much for the review suggestions, which are very helpful for my later research work. I have followed the suggestion to add a "Result and discussion" section to the article, which is used to describe the related work of this study, and at the same time, I have improved and modified the description of the experimental results and analysis in the "Experimental setup", and increased the details of the presentation.

11.After the result section, add a paragraph to clearly explain how your study represents incremental advance over previously published work.

Response: First of all, I would like to thank you very much for the review suggestions, which are very helpful for my later research work. As suggested, I have added a "Results and discussion" section after the results section in the article to explain that the present study is based on previously published research work and is an improvement of the work based on previous research.

12.The limitations of the work done should be discussed in the conclusion section. Discuss the major limitations of the study.

Response: First of all, I am very thankful for the review suggestions given, which will be very helpful for my future research work. As per the suggestions, I have discussed the limitations of the research work in this paper in the conclusion section of the article. That is, "This paper also faces some limitations and difficulties in the process of research work, mainly due to the fact that the multimodal remote sensing data sources are relatively few and difficult to obtain, and most of them are small samples of data, which is a limitation for a larger research work. Meanwhile, the proposed CMR-Net method involves some parameter tuning, which requires a lot of computation time during the training process and a better performing GPU card, which limits the fast presentation of the computational results."

13.Linguistic/grammatical/typographical error should be corrected before publication.

Response: First of all, I am very grateful for the advice given, I have scrutinized the article for some grammatical and typographical errors, and the problems found have been revised, and will be completed at a later stage before publication as required.

14.The paragraphs are not numbered, which should be done so comments and suggestions can be pointed out easily.

Response: Thank you very much for suggesting the review, it will be very helpful for my later research. Based on the suggestions I have revised and reformatted the paper according to the recommended formatting guidelines of the journal.

10. Chapter arrangements, numbering and formatting should be corrected in line with this journal’s recommended formatting guidelines.

Response: Suggestions for review are greatly appreciated and have been revised and reformatted in accordance with the journal's recommended formatting guidelines.

15.The paper does not discuss any potential drawbacks or shortcomings of the CMR-Net method in comparison to other state-of-the-art algorithms. Hence, the authors should provide a detailed discussion on the limitations and challenges faced during the implementation of the CMR-Net architecture, as this would help in understanding the potential drawbacks of the proposed method. 

Response: Thank you very much for the review suggestion, which is very helpful for my later research work. I have suggested that the limitations and challenges faced during the implementation of the CMR-Net method are discussed in detail in the "Result and discussion" section of the article, and also pointed out that the method increases the number of CNN convolutional layers and spends more computational overheads and does not provide the best results. It is also pointed out that the method increases the number of layers of CNN convolutional layers and costs more computational overhead and is not validated by fusing data from more data sources (e.g., three data sources).

16.Conduct a comparative analysis with a wider range of state-of-the-art algorithms for multi-modality remote sensing data classification, to provide a more comprehensive evaluation of the performance of CMR-Net.

Response: Thank you very much for the review suggestions, for my later research work is very helpful, I have been in accordance with the suggestions in the article in the "Experimental setup" part of the comparative analysis with the state-of-the-art algorithms in the introduction of the latest literature put forward EndNet method, as well as other deep learning methods for performance comparison, classification accuracy comparison and model training. The performance comparison, classification accuracy comparison and model training overhead are analyzed.

17.Include a discussion on the computational cost and efficiency of the CMR-Net method, as increasing the number of layers in the network may impact the convergence speed and computational requirements. 

Response: Thank you very much for the review suggestions, which are very helpful for my later research work. I have discussed the computational cost and efficiency of the CMR-Net method due to the increase of the number of layers in the network in the "Result and discussion" part of the article according to the suggestions, which pointed out that in the implementation of the CMR-Net classification method, the number of layers in the CNN convolutional layer is increased and more computational overhead is spent. It is discussed that the CMR-Net classification method is implemented by increasing the number of CNN convolutional layers and incurring more computational overhead in order to extract multimodal features more adequately, and the method performs cross-modal feature fusion on the extracted features. It is more accurate compared to the other two deep learning methods, but it also increases the computational cost and the network convergence will be slower. This method of cross-modal feature fusion of the extracted features also increases the computational cost and the network convergence will be slower compared to the other two deep learning methods in the paper. It is observed experimentally that the computational overhead of CMR-Net is about 10 minutes more than EndNet. However, this can be changed by improving the performance of the GPU compute card.

---

## [Decision Letter · Decision Letter 1]

22 Nov 2023

PONE-D-23-24147R1CMR-Net: A Cross Modality Reconstruction Network for Multi-modality Remote Sensing ClassificationPLOS ONE

Dear Dr. Wang,

Thank you for submitting your manuscript to PLOS ONE. After careful consideration, we feel that it has merit but does not fully meet PLOS ONE’s publication criteria as it currently stands. Therefore, we invite you to submit a revised version of the manuscript that addresses the points raised during the review process.

We look forward to receiving your revised manuscript.

Kind regards,

Ashwani Kumar, Ph.D.

Academic Editor

PLOS ONE

Journal Requirements:

Reviewers' comments:

Reviewer's Responses to Questions

**Comments to the Author**

1. If the authors have adequately addressed your comments raised in a previous round of review and you feel that this manuscript is now acceptable for publication, you may indicate that here to bypass the “Comments to the Author” section, enter your conflict of interest statement in the “Confidential to Editor” section, and submit your "Accept" recommendation.

Reviewer #1: All comments have been addressed

Reviewer #4: (No Response)

2. Is the manuscript technically sound, and do the data support the conclusions?

Reviewer #1: Yes

Reviewer #4: Yes

3. Has the statistical analysis been performed appropriately and rigorously? 

Reviewer #1: Yes

Reviewer #4: Yes

4. Have the authors made all data underlying the findings in their manuscript fully available?

Reviewer #1: Yes

Reviewer #4: No

5. Is the manuscript presented in an intelligible fashion and written in standard English?

Reviewer #1: Yes

Reviewer #4: Yes

6. Review Comments to the Author

Reviewer #1: (No Response)

Reviewer #4: The author(s) have made good efforts to incorporate the comments, the added parts, figures and tables add clarity to the methodology opted as well as the results. However, I feel there are still some points that need to be implemented for better understanding for the upcoming researchers and readers. Thus, I would advise the following recommendations to be taken into consideration by the author(s) before submitting the next revision-

1. In reference to comment 1, the revised ‘Introduction’ is inspiring and well written but it still does not elaborate the framework satisfactorily. Although, the author(s) have tried to improve the section by mentioning the contribution of the study in points but it still looks scattered at some places. Thus, it is suggested to update the part with the comprehensive background for better understanding.

2. In reference to comment 2, it is again suggested to add a separate paragraph at the end of introduction to justify and give the significance of your study clearly.

3. As per comment 4, the author(s) have improved the mathematical foundation of the method, however, it is also advised to explain the evaluation metrics used in the study and give a brief justification mentioning the reason to use them.

4. The author(s) are also suggested to exclude the irrelevant references from the ‘References’ section.

5. The author(s) should proof read the manuscript once again or get it done by an expert and correct all the linguistic mistakes and grammatical as well as typographical errors.

7. PLOS authors have the option to publish the peer review history of their article (what does this mean?). If published, this will include your full peer review and any attached files.

Reviewer #1: No

Reviewer #4: No

---

## [Author Response · Author response to Decision Letter 1]

25 Nov 2023

1. the revised ‘Introduction’ is inspiring and well written but it still does not elaborate the framework satisfactorily. Although, the author(s) have tried to improve the section by mentioning the contribution of the study in points but it still looks scattered at some places. Thus, it is suggested to update the part with the comprehensive background for better understanding.

Response: Thank you very much for the reviewer suggestions, which are very helpful for my later research work. I adjusted the "Introduction" according to the reviewer's suggestion, updated the integrated background introduction of this part, and clearly expressed the relevant background introduction of this research.

2. it is again suggested to add a separate paragraph at the end of introduction to justify and give the significance of your study clearly.

Response: First of all, I would like to thank you very much for the reviewer suggestions, which are very helpful for my later research work. In accordance with the reviewer's suggestion, I added a paragraph at the end of the "Introduction" to introduce the significance of this research, clearly explaining the purpose and significance of this paper.

3. the author(s) have improved the mathematical foundation of the method; however, it is also advised to explain the evaluation metrics used in the study and give a brief justification mentioning the reason to use them.

Response: First of all, I would like to thank you very much for the review suggestions, which are very helpful for my later research work. As suggested by the reviewers, I have added the reasons and index contents of the three commonly used evaluation indicators, Overall Accuracy (OA), Average Accuracy (AA), and kappa coefficient (κ), to the section of “Evaluation metric”.

4. The author(s) are also suggested to exclude the irrelevant references from the ‘References’ section.

 Response: First of all, I would like to thank you very much for the review suggestions, which are very helpful for my later research work. According to the suggestions of the reviewers, I have rearranged the references, and the references not used in this paper have been removed. The results are not suitably described. It is suggested to put in the details and strengthen the same.

5. The author(s) should proof read the manuscript once again or get it done by an expert and correct all the linguistic mistakes and grammatical as well as typographical errors.

Response: First of all, I would like to thank you very much for the review suggestions, which are very helpful for my later research work. As suggested by the reviewer, I have read the article carefully and revised the grammatical errors and language errors. Such as "The structure of this paper is as follows:" in the last paragraph of the introduction.

---

## [Decision Letter · Decision Letter 2]

17 May 2024

PONE-D-23-24147R2CMR-Net: A Cross Modality Reconstruction Network for Multi-modality Remote Sensing ClassificationPLOS ONE

Dear Dr. Wang,

Thank you for submitting your manuscript to PLOS ONE. After careful consideration, we feel that it has merit but does not fully meet PLOS ONE’s publication criteria as it currently stands. Therefore, we invite you to submit a revised version of the manuscript that addresses the points raised during the review process.

We look forward to receiving your revised manuscript.

Kind regards,

Muhammad Mateen

Academic Editor

PLOS ONE

Journal Requirements:

Additional Editor Comments:

Please revise the manuscript version 2 (latest version) according to the following comments.

1. In the start of Introduction section: "Hyperspectral image (HS), multispectral image (MS)"     It is better to write as: Hyperspectral (HS) image, multispectral (MS) image

2. "With the advancement of aerospace technology, Remote Sensing (RS) has emerged as a pivotal tool in Earth observation."     It is better to replace "Remote Sensing (RS)" with remote sensing (RS). and afterward use RS instead of the full term, Check the same in the whole manuscript to make it standard.

3. Same issue in "Index Terms-Classification, convolutional neural networks (CNNs), remote sensing (RS), Cross Modality Reconstruction (CMR), deep learning (DL), light detection and ranging (LiDAR), synthetic aperture radar (SAR)." It is better to keep the standard starting words with small letters and replace with;   cross modality reconstruction (CMR) .

4. Subheading of Results and discussion, "Data Description"  It is better to write as Data description.

5. Replace "berlin" with Berlin and make it correct.

6. It is recommended to cite few relevant references of year 2023 and 2024.

Reviewers' comments:

Reviewer's Responses to Questions

**Comments to the Author**

1. If the authors have adequately addressed your comments raised in a previous round of review and you feel that this manuscript is now acceptable for publication, you may indicate that here to bypass the “Comments to the Author” section, enter your conflict of interest statement in the “Confidential to Editor” section, and submit your "Accept" recommendation.

Reviewer #1: All comments have been addressed

Reviewer #5: All comments have been addressed

2. Is the manuscript technically sound, and do the data support the conclusions?

Reviewer #1: Yes

Reviewer #5: Yes

3. Has the statistical analysis been performed appropriately and rigorously? 

Reviewer #1: Yes

Reviewer #5: Yes

4. Have the authors made all data underlying the findings in their manuscript fully available?

Reviewer #1: Yes

Reviewer #5: Yes

5. Is the manuscript presented in an intelligible fashion and written in standard English?

Reviewer #1: Yes

Reviewer #5: Yes

6. Review Comments to the Author

Reviewer #1: (No Response)

Reviewer #5: I recommend to accept the article for publication in PLOS ONE journal

.....................

7. PLOS authors have the option to publish the peer review history of their article (what does this mean?). If published, this will include your full peer review and any attached files.

Reviewer #1: No

Reviewer #5: No

---

## [Author Response · Author response to Decision Letter 2]

21 May 2024

1. In the start of Introduction section: "Hyperspectral image (HS), multispectral image (MS)" , It is better to write as: Hyperspectral (HS) image, multispectral (MS) image.

Response: Thanks to the editor and reviewers for their comments, I have changed "Hyperspectral image (HS), multispectral image (MS)" to "Hyperspectral (HS) image, multispectral (MS) image" in the introduction. ".

2. "With the advancement of aerospace technology, Remote Sensing (RS) has emerged as a pivotal tool in Earth observation." It is better to replace "Remote Sensing (RS)" with remote sensing (RS). and afterward use RS instead of the full term, Check the same in the whole manuscript to make it standard.

Response: Thanks to the comments of the editor and reviewers, I have double-checked the standardisation of "Remote Sensing (RS)" throughout the article and used the term "RS" instead of "remote sensing" in the article.

3. Same issue in "Index Terms-Classification, convolutional neural networks (CNNs), remote sensing (RS), Cross Modality Reconstruction (CMR), deep learning (DL), light detection and ranging (LiDAR), synthetic aperture radar (SAR)." It is better to keep the standard starting words with small letters and replace with; cross modality reconstruction (CMR).

Response: Thanks to the comments made by the editor and reviewers, I have checked and modified the initial letters of the words in the index terms to lower case letters. The modification is: "Index Terms-classification, convolutional neural networks (CNN), remote sensing, cross modality reconstruction (CMR), deep learning (DL), light detection and ranging (LiDAR), synthetic aperture radar (SAR) light detection and ranging (LiDAR), synthetic aperture radar (SAR).".

4. Subheading of Results and discussion, "Data Description" It is better to write as Data description.

Response: Thanks to the editor and reviewers for their comments, I have changed the subtitle of Results and Discussion from "Data Description" to "Data description".

5. Replace "berlin" with Berlin and make it correct.

Response: Thanks to the editor and the reviewers for their comments, I have checked and corrected "berlin" to "Berlin" in the article.

6. It is recommended to cite few relevant references of year 2023 and 2024.

Response: Thanks to the suggestions from the editors and reviewers, I have cited relevant literature from 2023 and 2024 in the "Related work" of the article, and modified the order and format of some of the literature in the "References". The references cited in 2023 and 2024 are as follows: 

[30] F.M. Guo, Z.W. Li, et al, "Semi-supervised cross-domain feature fusion classification network for coastal wetland classification with hyperspectral and LiDAR data," INT J APPL EARTH OBS, vol. 120, 2023.

[31] Y.N. Feng, J.H. Zhang, et al. "S2EFT: Spectral-Spatial-Elevation Fusion Transformer for hyperspectral image and LiDAR classification," KNOWL-BASED SYST, vol. 283, 2024.

[33] 33. Xin. He, Y.S. Chen, et al, "Foundation Model-Based Multimodal Remote Sensing Data Classification. " IEEE Trans. Geosci. Remote Sens. vol. 62, 2024.

---

## [Editor Report · Decision Letter 3]

22 May 2024

CMR-Net: A Cross Modality Reconstruction Network for Multi-modality Remote Sensing Classification

PONE-D-23-24147R3

Dear Dr. Wang,

We’re pleased to inform you that your manuscript has been judged scientifically suitable for publication and will be formally accepted for publication once it meets all outstanding technical requirements.

Kind regards,

Muhammad Mateen

Academic Editor

PLOS ONE

Additional Editor Comments (optional):

All the comments have been addressed by authors
---

## [Editor Report · Acceptance letter]

14 Jun 2024

PONE-D-23-24147R3 

PLOS ONE

Dear Dr. Wang, 

I'm pleased to inform you that your manuscript has been deemed suitable for publication in PLOS ONE. Congratulations! Your manuscript is now being handed over to our production team.

Kind regards, 

on behalf of

Dr. Muhammad Mateen 

Academic Editor

PLOS ONE